# miR-410 Is a Key Regulator of Epithelial-to-Mesenchymal Transition with Biphasic Role in Prostate Cancer

**DOI:** 10.3390/cancers16010048

**Published:** 2023-12-21

**Authors:** Diana M. Asante, Amritha Sreekumar, Sandip Nathani, Tae Jin Lee, Ashok Sharma, Nikhil Patel, Matthew N. Simmons, Sharanjot Saini

**Affiliations:** 1Department of Biochemistry and Molecular Biology, Augusta University, Augusta, GA 30912, USA; dasante@augusta.edu (D.M.A.); amsreekumar@augusta.edu (A.S.); snathani@augusta.edu (S.N.); 2Department of Center for Biotechnology and Genomic Medicine, Augusta University, Augusta, GA 30912, USA; talee@augusta.edu (T.J.L.); assharma@augusta.edu (A.S.); 3Department of Pathology, Augusta University, Augusta, GA 30912, USA; naptel4@augusta.edu; 4Department of Urology, Augusta University, Augusta, GA 30912, USA; matsimmons@augusta.edu

**Keywords:** miR-410, prostate cancer, EMT, neuronal markers

## Abstract

**Simple Summary:**

The molecular basis of prostate cancer progression to advanced disease is not fully understood. Small non-coding RNAs, called microRNAs regulate gene expression post-translationally and regulate crucial cellular processes. It has been reported that microRNAs are dysregulated during prostate cancer progression. The objective of this study is to determine the role of miR-410-3p, a little-studied microRNA in prostate cancer progression. Our results demonstrate for the first time that this miRNA plays a context-dependent role in prostate cancer. We found that miR-410 expression is regulated in a biphasic manner in prostate cancer, with its expression decreased in primary prostate cancer and increased in the advanced disease. We further performed functional studies that show that miR-410 regulates critical cellular processes that are involved in cancer progression and metastasis in prostate cancer. Our study suggests that miR-410-mediated regulation of target genes is context-dependent. Our findings would lead to an increased understanding of the mechanistic basis of prostate cancer progression and metastasis that would be helpful in guiding novel therapies.

**Abstract:**

The molecular basis of prostate cancer (PCa) progression from the primary disease to metastatic castration-resistant prostate cancer (CRPC) followed by therapy-induced neuroendocrine prostate cancer is not fully understood. In this study, we elucidate the role of miR-410, a little-studied microRNA located on chromosome 14q32.31 within the DLK1-DIO3 cluster, in PCa. miR-410 expression analyses in primary and metastatic PCa tissues and cell lines show that its levels are decreased in initial stages and increased in advanced PCa. Functional studies were performed in a series of PCa cell lines. In LNCaP cells, miR-410 overexpression led to decreases in cellular viability, proliferation, invasiveness, and migration. On the other hand, miR-410 overexpression in PC3 and C42B cells led to increased viability, proliferation, and invasiveness. Our data suggest that miR-410 represses epithelial-to-mesenchymal transition (EMT) in LNCaP cells by directly repressing SNAIL. However, it promotes EMT and upregulates PI3K/Akt signaling in PC3 and C42B cells. In vivo studies with PC3 xenografts support an oncogenic role of miR-410. These data suggest that miR-410 acts as a tumor suppressor in the initial stages of PCa and play an oncogenic role in advanced PCa. Our findings have important implications in understanding the molecular basis of PCa progression with potential translational implications.

## 1. Introduction

Prostate cancer (PCa) is a leading cause of male cancer-related fatalities in the United States (US) with an estimated 34,700 deaths projected for 2023 [1]. This cancer is primarily driven by androgens acting through the androgen receptor signaling pathway. The primary approach to prostate cancer therapy involves androgen deprivation or AR inhibition [2] leading to an initial regression of the disease. Nevertheless, about 2–3 years after anti-androgen therapy initiation, the condition progresses into castration-resistant prostate cancer (CRPC), which presents limited treatment options and poor survival rates [3]. CRPC often manifests with metastatic spread, commonly to the bones, giving rise to osteoblastic and osteolytic lesions [4]. The next generation of therapies for CRPC include AR pathway inhibitors such as enzalutamide (MDV3100/ENZ) and abiraterone. These drugs are employed for men with both metastatic and non-metastatic CRPC and initially contribute to improved survival. However, CRPC patients eventually develop drug resistance over time, attributable to heterogeneous molecular mechanisms that are either AR dependent or AR independent [5]. AR dependent mechanisms entail AR amplification, AR point mutations, emergence of ligand-independent splice variants, and alterations in intratumoral androgen biosynthesis. AR independent resistance mechanisms on the other hand encompass AR bypass signaling, immune-mediated resistance mechanisms or neuroendocrine differentiation (NED) [6,7]. The latter refers to the trans-differentiation of CRPC-adenocarcinomas to the neuroendocrine variant, referred to as neuroendocrine prostate cancer (NEPC). As a result of NED, PCa cells exhibit a reduced expression of luminal lineage markers such as AR and PSA coupled with increased expression of alternative neuroendocrine (NE) lineage markers such as enolase 2 (ENO2), chromogranin A (CHGA) and synaptophysin (SYP) [8,9]. Due to the absence of AR signaling, these PCa variants become resistant to anti-androgen therapy and constitute an extremely aggressive variant of advanced CRPC, characterized by shorter survival times (<1 year) and limited therapeutic options. These variants are often linked to the presence of visceral metastasis to the liver, lung and central nervous system, in addition to lytic bone metastases and low serum PSA levels relative to the extent of the disease burden [8]. The precise molecular underpinnings of the progression of PCa from primary disease to metastatic CRPC and ultimately therapy-induced NEPC remain incompletely understood.

A series of genetic alterations and epigenetic changes promote PCa progression and metastasis [10]. Prominent epigenetic regulators of PCa progression are non-coding RNAs, including microRNAs [11]. MicroRNAs (miRNAs) are small, endogenous non-coding RNAs that suppress gene expression post transcriptionally via sequence-specific interactions with the 3′-untranslated regions (UTRs) of cognate mRNA targets [12]. These miRNAs play an integral role in a wide spectrum of cellular functions such as proliferation, apoptosis and cellular differentiation [13]. Since these cellular functions are altered in cancer, miRNAs are often dysregulated in cancer. In prostate cancer, miRNAs have been shown to play important regulatory roles in cancer initiation, progression and metastasis [10]. miRNA levels are dysregulated by varying mechanisms such as methylation of their promoters or by other tumor promoting factors or by genomic loss or deletion [14]. Based on their gain and loss of function, miRNAs are classified as tumor-suppressors (loss of expression) or oncogenes (over-expressed). Typically, tumor suppressor miRNAs repress oncogenes such as *MYCN* and *AURKA* [15], while oncogenic miRNAs repress the expression of tumor suppressor genes such as *TP53*, *PTEN* and *RB1*. However, these functions are known to be context-dependent and may vary from one cell type to another [16].

In this study, we examined the role of miR-410-3p (referred to as miR-410) in prostate cancer. This miRNA is located on chromosome 14q32.31 in the DLK1-DIO3 (delta-like 1 homolog-deiodinase iodothyronine 3) cluster that is known to be activated in embryonic stem cells [17]. The role of miR-410 located within this cluster is not fully understood in prostate cancer. One study identified miR-410 as an oncogenic microRNA in prostate cancer, demonstrating its direct repression of the tumor suppressor PTEN [18]. In other cancers, miR-410 has been shown to play an oncogenic or tumor suppressor role, depending upon the context. In non-small-cell lung cancer, miR-410 was shown to inhibit GSK-3β and induce its degradation, promoting the cancer stem cell (CSC) phenotype. β-catenin was found to be upregulated by miR-410 in this study [19].

Our laboratory is interested in delineating miRNAs with roles in advanced prostate cancer. Our preliminary screening identified that miR-410 expression is dysregulated in this disease. We sought to examine its role in further detail. We demonstrate to our knowledge for the first time that miR-410 plays a biphasic role in PCa. Our study suggests that miR-410 acts as a tumor suppressor in the initial stages of the disease, wherein it represses epithelial-to-mesenchymal transition (EMT) by directly repressing EMT inducing transcription factor, SNAIL. However, it exerts an oncogenic role in higher stage and grade of the disease by inhibiting protein kinase D1 (PRKD1), a serine/threonine kinase in advanced PCa. Our findings have important implications in understanding the molecular basis of PCa progression with translational potential.

## 2. Materials and Methods

### 2.1. Cell Lines and Cell Culture

The nonmalignant prostate epithelial cell line RWPE-1 along with various prostate carcinoma cell lines (LNCaP, C42B, PC3, Du145, NCI-H660) were procured from the American Type Culture Collection (ATCC, Manassas, VA, USA) and cultured under the recommended conditions. All these cell lines were consistently maintained in an incubator with a humidified atmosphere of 95% air and 5% CO_2_ at a temperature of 37 °C. The identity of these prostate cell lines was confirmed via DNA short-tandem repeat analysis. The experiments involving these cell lines were conducted within 6 months of their acquisition/resuscitation. 

### 2.2. miRNA Transfections

At 24 h prior to transfection, cells were plated in growth medium without antibiotics. Transient transfections were conducted using miRNA precursors (Thermo Fisher Scientific, Waltham, MA, USA) by using lipofectamine 2000 (Thermo Fisher Scientific, Waltham, MA, USA) according to the manufacturer’s protocol. Control miRNA (miR-CON; AM17110)/miR-410-3p precursor (Catolog no. 4464084) was used for miRNA transfections followed by functional assays. All miRNA transfections were for 72 h. For PC3 and C42B cells, miR-410/scrambled miR-410 plasmid was stably transfected using lipofectamine 2000 followed by selection in 1 µg/mL of puromycin. pMIR-hsa-mir-410-GFP, miRNA vector (pMIR) with CMV promoter-driven expression of human pre-miRNA hsa-mir-410 and GFP reporter and the corresponding scrambled control was obtained from Vigene Biosciences (Rockville, MD, USA).

### 2.3. Cell Viability Assays

Cell viability was assessed at 24, 48, 72 h following miR-CON/miR-410 transient transfections using the CellTiter 96 AQueousOne Solution Cell Proliferation Assay Kit (Promega, Madison, WI, USA), according to the manufacturer’s protocol.

### 2.4. Cell Cycle Analyses

To analyze the cell cycle using fluorescence-activated cell-sorting (FACS), 72 h post-transfection, cells were harvested, washed with cold PBS, and resuspended in propidium iodide (PI) stain (BD Biosciences, Franklin Lakes, NJ, USA) in accordance with the manufacturer’s guidelines. Subsequently, the stained cells were immediately analyzed by FACS (Cytoflex; Beckman Coulter, Inc., Brea, CA, USA).

### 2.5. Clonogenicity Assay

After 72 h post-transfection, the cells were counted, seeded at a low density of 1000 cells/plate and allowed to grow until visible colonies became evident. Then, cells were subjected to Giemsa staining, and the count of colonies was conducted.

### 2.6. Migration and Invasion Assays

Following transient transfections, cell counting was conducted 48 h post-transfection, after which transwell migration and invasion assays were executed following the manufacturer’s instructions. The cells were seeded onto control inserts or matrigel inserts (BD Biosciences, Franklin Lakes, NJ, USA) at a concentration of 1 × 10^5^ cells per milliliter in serum-free media, and they were allowed to migrate or invade for 24 h at 37 °C. Subsequently, cells on the top side of the inserts were removed, and those that had migrated or invaded through the polycarbonate or basement membrane were fixed, stained, and enumerated under a microscope (Keyence, Itasca, IL, USA).

### 2.7. Western Blotting

Whole-cell extracts were prepared in RIPA buffer, which consisted of 50 mmol/L Tris (pH 8.0), 150 mmol/L NaCl, 0.5% deoxycholate, 0.1% SDS, and 1.0% NP-40, and included a protease inhibitor cocktail from Roche. Total protein was separated through SDS-PAGE electrophoresis, and Western blotting was performed following established protocols. The antibodies for Western blotting were procured from Cell Signaling (Danvers, MA, USA). Following antibodies were employed: E-cadherin (Catalog no. 3195), Vimentin (Catalog no. 5741), N-cadherin (Catalog no. 4061), SNAIL (Catalog no. C15D3), MMP10 (Catalog no. 13130), GAPDH (Catalog no. 2118). The original western blot figures can be found in Appendix A.

### 2.8. Xenograft Tumors

Animal experiments were conducted in strict adherence to the institutional guidelines of Augusta University under an approved protocol. PC3 cells that stably overexpressed either control miRNA or miR-410 were subcutaneously injected into nude mice (4–5-week-old, Jackson Laboratories, *n* = 6 for control, *n* = 8 for miR-410). The cell injections were administered into the right or left flanks of the mice in a 100 µL volume mixed with 50% matrigel. Once palpable tumors became visible, caliper measurements were taken on a weekly basis, and tumor volumes were computed using the formula x^2y/2^, where x < y.

### 2.9. RNA and miRNA Extraction

Total RNA was extracted from microdissected FFPE tissues and cultured cells using a miRNeasy FFPE Kit (Qiagen, Hilden, Germany) and an miRNeasy mini kit (Qiagen, Hilden, Germany), respectively, following the manufacturer’s instructions.

### 2.10. Quantitative Real-Time PCR

Mature miRNAs and mRNAs were assayed using the TaqMan MicroRNA Assays and Gene Expression Assays, respectively, following the manufacturer’s instructions (Thermo Fisher Scientific, Waltham, MA, USA). The expressions of miRNAs and mRNAs were normalized using RNU48 or GAPDH as control references, respectively. The specific Taqman microRNA assays employed included hsa-miR-410-3p (assay ID 4427975) and RNU48 (assay ID 001006). Gene expression assays employed were SYP (Hs00300531_m1), ENO2 (Hs00157360_m1), N-Myc (Hs00232074_m1), BRN2 (Hs00271595_s1) and GAPDH (Hs99999905_m1). Relative changes in gene expression were calculated using the comparative Ct method on the StepOnePlus Real Time PCR System (Thermo Fisher Scientific, Waltham, MA, USA).

### 2.11. Luciferase Assays

For SNAIL, the two 3′ UTR binding sites for miR-410, denoted as SNAIL-1 and SNAIL-2, were cloned into pmiR-Glo luciferase reporter vector. Control pmiR-Glo and SNAIL 3′ UTR constructs (0.2 µg) were transiently transfected into cells stably transfected with control/miR-410 using lipofectamine 2000 (Thermo Fisher Scientific, Waltham, MA, USA). After 48 h following transfection, we measured Firefly and Renilla luciferase activities using the dual luciferase reporter assay system (Promega, Madison, WI, USA) according to the manufacturer’s instructions. Firefly luciferase activity was normalized to Renilla luciferase activity for each sample and relative luciferase activities were calculated.

### 2.12. Microarray Analyses

Microarray analyses were performed in C42B and PC3 cells stably transfected with miR-CON/miR-410 using a Clariom S platform at Augusta University Molecular Core Facility. Total RNA (1–2 µg) was used for sample preparation. RNA quality was assessed by the Agilent 2200 TapeStation (Agilent Technologies, Santa Clara, CA, USA) ensuring an RNA Integrity Number (RIN) ≥ 7. Total RNA samples were processed using the GeneChip WT PLUS Reagent Kit (Thermo Fisher Scientific, Waltham, MA, USA). Briefly, 250 ng of total RNA was used to generate sense strand cDNAs. These sense strand cDNAs were subsequently fragmented and labeled with biotin as per manufacturer’s instructions. The labeled samples were then subjected to hybridization on the Clariom S human array (Thermo Fisher Scientific, Waltham, MA, USA) that contains more than 22,100 well-annotated genes and 150,300 transcripts. Following a 16 h hybridization period, the arrays were washed and stained using Affymetrix GeneChip Fluidics Station 450 systems. Subsequently, the stained arrays were scanned on an Affymetrix GeneChip Scanner 3000, generating data in the form of CEL files. These CEL files were imported into Partek Genomic Suites version 6.6 (Partek, Chesterfield, MI, USA) using the standard import tool with a Robust Multi-array Average (RMA) normalization. Differential expressions were determined using ANOVA within the Partek package and filtered with a *p*-value cutoff of 0.05 and fold-change cutoffs to identify differentially expressed genes in each comparison. The list of significant genes was employed to create a hierarchical clustering plot based on standardized expression values. For the analysis and visualization of global expression patterns in genes, R version 4.2.2 was used. The difference in gene profile between two groups were obtained in form of fold changes. The fold changes were log transformed and gene set enrichment (GSEA) analysis was performed against gene ontology terms utilizing clusterProfiler package.

### 2.13. Statistics

All quantified data represent an average of triplicate samples or as indicated. Data are represented as mean ± S.E.M or as indicated. Statistical analyses were performed using Graphpad Prism 9.5.1.733. Results were considered statistically significant at *p* ≤ 0.05.

## 3. Results

### 3.1. miR-410 Expression Is Biphasic in Prostate Cancer

We examined miR-410 expression in primary PCa (Figure 1A). Prostate cancer clinical tissues (*n* = 9) were micro dissected and miR-410 expression was analyzed by real-time PCR in matched tumor/normal tissues (Figure 1A). miR-410 expression analyses showed that 8/9 tissues show a downregulated expression of this miRNA. We further probed the Cancer Genome Atlas (TCGA) dataset of prostate adenocarcinomas for miR-410 expression using UALCAN portal [20,21] (Figure 1B). Our analyses showed that as compared to normal tissues (*n* = 51), primary prostate adenocarcinomas (*n* = 490) show a significant attenuation of expression of this miRNA (*p* = 0.003 *). We further examined if miR-410 expression correlated with the Gleason score of primary prostate cancer. Our analyses showed that while in a Gleason score of 6–8, miR-410 expression was low, a Gleason score of >8 tended towards higher expression (Figure 1C). We also examined the expression of miR-410 in prostate adenocarcinomas based on their tumor nodal status (Figure 1D). While in N0 tissues, the expression of miR-410 was low, N1 tissues had a higher expression of miR-410. We further examined miR-410 expression across various subtypes of prostate adenocarcinomas. We found that miR-410 expression depends upon the tumor subtype, with those with ERG fusion, ETV1 fusion and SPOP mutations having a lower miR-410 expression as compared to other subtypes (Figure 1E). miR-410 expression analyses in PCa cell lines (Figure 1F) showed that as compared to normal immortalized prostate epithelial cell line RWPE-1, PCa cell lines LNCaP and Du145 had a lower miR-410 expression, while levels in C42B were not much altered. However, PC3 cells and NEPC cell line NCI-H660 were shown to have higher miR-410 expression as compared to RWPE-1 cells. We further analyzed miR-410 expression in microdissected CRPC and NEPC clinical samples (Figure 1G) by real-time PCR analyses. Our data show that miR-410-3p expression tends towards increased levels in NEPC. These data suggest that miR-410 expression is biphasic in prostate cancer, with its levels decreased in the initial stages and increasing in the advanced stage disease. We further examined if miR-410 expression is regulated by androgens. To examine this, we treated LNCaP cells with dihydrotestosterone (DHT) followed by real-time PCR based analyses (Appendix A). We found that miR-410 is significantly downregulated by DHT treatment as compared to control cells.

### 3.2. miR-410 Regulates Proliferation of PCa Cells in a Context Dependent Manner

In view of the observed biphasic expression data of miR-410 in clinical tissues and cell lines, we sought to examine the functional role of this miRNA in prostate cancer. We hypothesized that this miRNA plays a context dependent role in prostate cancer. To examine our hypothesis, we sought to perform functional studies on this miRNA in multiple PCa lines. We overexpressed miR-410-3p in LNCaP, C42B and PC3 (Figure 2A) cells, followed by functional assays. The assessment of miR-410 expression by real-time PCR confirmed its overexpression in miR-410 transfected cells as compared to control cells (Figure 2A). Analyses of cellular viability by MTS assay showed that miR-410 exerts opposite effects in PCa cell lines, with cellular viability decreasing in LNCaP cells and increasing in C42B and PC3 cells over days 2, 3 and 4 as compared to their respective controls (Figure 2B). We further assayed the clonogenicity potential of miR-410 transfectants (Figure 2C) that showed that in LNCaP cells, the colony formation ability is lower as compared to the control, whereas in PC3 and C42B, colony numbers were increased as compared to the corresponding controls.

### 3.3. miR-410 Regulates Cell Cycle Progression of Prostate Cancer Cells in a Context-Dependent Manner

We further assessed the effects of miR-410 expression on the cell cycle by FACS analyses of propidium iodide (PI) stained cells (Figure 3). miR-410 expression in LNCaP cells caused an increase in the G0-G1 phase of the cell cycle with a concomitant decrease in the G2-M phase as compared to miR-CON transfected cells (Figure 3A). However, miR-410 expression in C42B and PC3 cells caused an increase in the S-phase as compared to their corresponding controls (Figure 3B,C). These data suggest a potential tumor suppressor role of miR-410 in androgen-dependent LNCaP cells and an oncogenic role in androgen-independent PC3 and C42B cell lines.

### 3.4. miR-410 Overexpression Influences Migratory and Invasive Properties of PCa Cell Lines

To examine the role of miR-410 in influencing in vitro invasiveness and the migratory ability of PCa cell lines, transwell invasion and migration assays were conducted (Figure 4). miR-410 overexpression led to decreased migratory and invasive abilities in LNCaP cells as compared to the control (Figure 4A). Overexpression of miR-410 in C42B (Figure 4B) and PC3 cells (Figure 4C) led to increased invasion and migration. These data support a context-dependent role of miR-410 as a tumor suppressor or oncogene in prostate cancer.

### 3.5. miR-410 Regulates Epithelial-to-Mesenchymal Transition in Prostate Cancer

In view of its potential effects on the EMT pathway, we analyzed the effects of miR-410 expression on epithelial markers and mesenchymal markers in PCa cell lines. Western blot analyses of LNCaP cells showed an upregulation of epithelial marker E-cadherin and downregulation of mesenchymal marker N-cadherin upon miR-410 expression as compared to the corresponding control, supporting an inhibitory effect of miR-410 on EMT in LNCaP cells (Figure 5A). These observations are in agreement with the potential tumor suppressor role of miR-410 in LNCaP cells. In view of its effects on EMT, we analyzed its potential to directly regulate EMT genes. In silico analyses with Targetscan [22] showed that SNAIL/SNAI1 is a potential miR-410 target as it has two potential miR-410 binding sites within its 3′ untranslated region (UTR), referred to as SNAI1-1 and SNAI1-2 (Figure 5B). Western blot analyses of SNAIL expression showed that miR-410 overexpression causes decreased SNAIL expression (Figure 5C). To validate SNAIL as a direct miR-410 target, we cloned its potential 3′ UTR binding sites 1 and 2 into pmiR-Glo (Promega) luciferase reporter vector. miR-CON/miR-410 was co-transfected with control/SNAI1-1/SNAI1-2 3′ UTR constructs into LNCaP cells followed by luciferase reporter assays (Figure 5D). This assay showed that miR-410 represses the activity of SNAIL via binding to site 2 in its 3′-UTR.

In view of these effects on EMT, we next asked if miR-410 inhibits EMT in C42B and PC3 cells as well. However, contrary to its effects on LNCaP cells, miR-410 overexpression led to a decrease in epithelial marker E-cadherin expression (Figure 5E) in C42B and PC3 cells. SNAIL expression was found to be upregulated in C42B cells, suggesting a context-dependent repression of SNAIL by miR-410 in PCa. SNAIL levels in PC3 cells were not much altered. Also, we found an upregulation of matrix metalloproteinase MMP10 in C42B and PC3 cells (Figure 5F), supporting a potential oncogenic role of this miRNA in these cell lines. In agreement with the tumor suppressor role of miR-410 in LNCaP cells, a decrease in MMP10 expression was observed (Figure 5F).

### 3.6. Microarray Analyses of miR-410 Target Genes Reveals Its Important Role in Critical Cellular Processes

To examine the potential target genes of miR-410 underlying its oncogenic role in PC3 and C42B cells, we performed microarray analyses on the Clariom S platform in these cells stably transfected with miR-410 vs. scrambled miR-410 control construct (Figure 6).

Microarray analyses of PC3 cells (Figure 6A) showed that stable expression of miR-410 leads to the upregulation of 919 genes and downregulation of 741 genes. Downregulated genes included Protein Kinase D1 (PRKD1) and dihydropyrimidinase-like 3 (DPYSL3), among others, while upregulated genes included CEACAM5, epithelial splicing regulatory protein 1 (ESRP1), solute carrier family 6 (amino acid transporter) member 14 (SLC6A14), tandem C2 domains, nuclear (TC2N) and podoplanin (PDPN).

In C42B cells, 365 genes were upregulated and 273 genes were downregulated upon miR-410 expression as compared to the scrambled control (Figure 6B). Top upregulated genes included insulin like growth factor binding protein 3 (IGFBP3), testis expressed 15 (TEX15), SYT4 (synaptotagmin IV), and downregulated genes included SLIT and NTRK like family, member 3 (SLITRK3), cerebellin 2 precursor (CBLN2) and butrylcholinesterase (BCHE) (Table 1). Gene set enrichment analyses (GSEA) in PC3 cells (Figure 6C and Appendix A) showed an enrichment of genes involved in chromosome organization and segregation, signaling receptor activator activity (Appendix A), receptor ligand activity (Appendix A) and leukocyte chemotaxis (Appendix A). Importantly, we observed a negative enrichment of genes involved in protein serine/threonine kinase activity (Appendix A). GSEA analyses in C42B cells (Figure 6D) showed the enrichment of cell growth (Appendix A), angiogenesis (Appendix A) and positive regulation of cell activation. Interestingly, immune-related pathways such as those involved in leukocyte cell–cell adhesion and adaptive immune response based on somatic recombination of immune receptors built from immunoglobulin superfamily domains were represented (Appendix A).

### 3.7. Protein Kinase D1 Is a Potential miR-410 Target in Prostate Cancer

In view of our analyses showing potential regulation of genes involved in protein serine/threonine kinase activity (Appendix A), we focused on this gene set. Our analyses showed that Protein Kinase D1, a gene found to be highly downregulated by miR-410 overexpression in PC3 cells, possess a potential miR-410 binding site (Figure 7A). To confirm the microarray data, we assessed *PRKD1* expression levels in miR-410 transfected PC3 and C42B cells by real-time PCR (Figure 7B). We found a significant downregulation of *PRKD1* levels in C42B and PC3 cells upon miR-410 overexpression as compared to control transfected cells. We also examined protein levels of PRKD1 upon miR-410 overexpression by Western blot analyses (Figure 7C). In agreement with real time PCR data, Western blot analyses showed a significant downregulation of PRKD1 expression in miR-410 treated cells as compared to miR-CON treated cells. These data validate PRKD1 as a potential miR-410 target. We further sought to validate additional genes identified to be dysregulated by microarray analyses. We found that Akt1 and pan-Akt are upregulated by miR-410 expression in PC3 cells. The levels of these proteins were not much altered in C42B cells. C42B cells showed a prominent upregulation of TGF-β protein levels upon miR-410 overexpression, consistent with its effects on EMT in this cell line. These data suggest an important role of miR-410 in signaling pathways related to cellular proliferation and EMT.

### 3.8. Effects of miR-410 Overexpression In Vivo

Next, we sought to examine the functional impact of miR-410 expression in vivo in a PCa xenograft mouse model (Figure 8). PC3 cells stably transfected with miR-CON/miR-410 expression construct (Figure 2A) were used for these studies. Control miR or miR-410 expressing cells were subcutaneously injected into two groups of nude mice (*n* = 6 for control group, *n* = 8 for miR-410). Our results show that miR-410 overexpression led to increased growth of xenograft tumors over time, validating its role as an oncogenic miRNA in PC3 cells (Figure 8A). We harvested tumors from control and test mice followed by real-time PCR-based analyses of miR-410 expression (Figure 8B). Our analyses confirmed miR-410 overexpression in the test mice as compared to the control mice. These data validate the in vivo oncogenic role of miR-410.

### 3.9. miR-410 Regulates the Expression of Neuronal Markers in Prostate Cancer

In view of increased expression of miR-410 observed in NEPC tissues, we next asked if miR-410 expression plays a role in driving NEPC. Towards this, we performed real-time PCR-based expression analyses of neuronal markers upon miR-410 overexpression in C42B (Figure 9). Our analyses shows that the expression of neuronal markers, primarily synaptophysin (SYP) is regulated by miR-410. Overexpression of miR-410 in C42B led to a significant increase in SYP, ENO2 and non-significant increases in N-Myc and BRN2 expression.

Real-time PCR-based analyses of indicated neuronal markers in C42B cells transfected with miR-CON/miR-410. GAPDH was used as an endogenous control.

## 4. Discussion

The mechanistic basis of PCa progression and metastasis is not well understood leading to the lack of effective diagnostic and prognostic biomarkers for the disease. Therapeutic strategies for castration-resistant prostate cancer are limited, and the emergence of neuroendocrine differentiation in CRPC patients as a result of treatment with second generation of AR pathway inhibitors poses significant clinical challenges. It has been recognized that in addition to genetic alterations, PCa progression involves alterations in critical epigenetic factors such as miRNAs [10]. However, there are still gaps in our understanding of the role of miRNAs in this disease. In this study, we delineate the role of miR-410, a little studied miRNA in PCa. To our knowledge, to date, there is only one study that examined the role of miR-410 in PCa and reported it to be an oncogenic miRNA that directly represses the tumor suppressor PTEN [18]. In contrast, our findings suggest a potential dual role of miR-410 in prostate cancer. Our data suggest that this miRNA is downregulated by androgens and acts as a tumor suppressor in the early stages of primary PCa. However, with increasing disease aggressiveness, miR-410 expression increases, and in the context of the advanced disease, it behaves as an oncogenic miRNA. Similar to our findings on a potential context dependent role of miR-410 as either a tumor suppressor or oncogene in PCa, it has been reported that miR-410 exhibits a context-dependent role in adenomas [23]. This study showed that it acts as an oncogenic miRNA in gonadotroph and corticotroph adenoma cells where it activates MAPK, PTEN/AKT and STAT3 signaling pathway. However, in somatotroph adenoma cells, it has been reported to suppress the activity of these signaling pathways and thereby play a tumor suppressor role.

Epithelial-to-mesenchymal transition (EMT) is a complex embryonic developmental program that is reactivated in cancer cells and is orchestrated by multiple signaling pathways by various inducers such as the Transforming Growth Factor-β (TGF-β) [10]. The process of EMT is defined by alterations in gene expression, involving the downregulation of epithelial genes, such as E-cadherin and the upregulation of mesenchymal genes such as vimentin and N-cadherin. It is a pivotal step in tumor invasion and metastasis [10] and it is orchestrated by several transcription factors (TFs) including SNAIL and ZEB families [24]. As a result of EMT, cancer cells become motile and invasive, detaching from primary tumors and colonizing distant sites. At distant sites, cells often undergo the reverse process, referred to as mesenchymal-to-epithelial transition (MET), reverting back to immotile epithelial states, leading to the development of secondary tumors [25]. In prostate cancer (PCa), EMT has been implicated in playing a crucial role, particularly in the context of metastatic disease [26,27,28,29]. Epithelial marker E-cadherin is commonly lost or shows reduced expression in prostate cancer [26,27], along with a gain of N-cadherin. These EMT related changes are associated with multiple end points of progression and mortality [30]. A large body of evidence suggests that miRNAs are crucial regulators of EMT [31] in prostate cancer and other cancers [32,33,34]. Prominent examples of EMT regulating miRNAs include miR-200 family and miR-205 that directly target EMT-TFs including ZEB1 and ZEB2 [32,35,36,37,38,39,40,41], amongst other targets [38]. We showed the role of miR-203 in inhibiting PCa EMT by directly targeting ZEB2 and BMI1, thereby regulating PCa invasion and metastasis [27,42]. In the present study, we found that miR-410 regulates PCa EMT in a context-dependent manner. In the androgen-dependent PCa cell line LNCaP, miR-410 directly represses SNAIL via binding to its target site within its 3′ UTR, thereby inhibiting EMT. In view of these data, we propose that the loss of miR-410 expression in primary PCa would lead to SNAIL upregulation, promoting EMT and disease progression.

Contrasting the tumor suppressive effects of miR-410 in LNCaP cells, this miRNA exhibits oncogenic effects in PC3 and C42B cells pointing to its context-dependent role in prostate cancer. Our data support a context-dependent regulation of its target genes by miR-410. SNAIL was not found to be inhibited in C42B and PC3 cells in agreement with the opposite effects of this miRNA in PCa EMT in these cell types. In other cancers, miR-410 has been shown to play either an oncogenic role or tumor-suppressive role pointing to the dual effects of this miRNA [43]. miR-410 acts as an oncogene in non-small-cell lung cancer, liver cancer and colorectal cancer, while it acts as a tumor suppressor in pancreatic cancer, bone cancer and gastric cancer. miR-410 negatively regulates Bcl2 antagonist killer 1 (BAK1), Centrin 3 (CETN3) and Bromodomain containing protein 7 (BRD7) to promote other cancers. However, miR-410 effectively targets c-MET and Angiotensin II Type 1 receptor (AGTR1) to suppress cancer [43]. It was also reported to target SNAIL in MDA-MB-231 cells [43].

Importantly, in this study, we identify PRKD1 as a novel miR-410 target gene in androgen-independent cell lines. PRKD1 belongs to the family of calcium calmodulin kinases, which has been reported to be reduced in metastatic prostate cancer as compared to the localized disease [44]. PRKD1 signaling plays an important role in the regulation of cellular proliferation, adhesion, EMT, invasion, migration, immune response and angiogenesis [44,45]. Interestingly, it has been reported that PRKD1 suppresses EMT through phosphorylation of SNAIL [46]. PRKD1 phosphorylates SNAIL on Serine 11 that triggers the nuclear export of SNAIL via binding to a 14-3-3σ protein. Upon nuclear export, phosphorylated SNAIL cannot repress E-cadherin effectively, leading to E-cadherin induction [46]. In light of these studies and our present results, we propose that the biphasic expression of miR-410 expression in PCa cell lines is inversely correlated to PRKD1 expression that determines PCa EMT outcomes. We propose that in advanced prostate cancer, miR-410 upregulation leads to PRKD1 inhibition that promotes EMT, likely via non-phosphorylation of SNAIL and other mechanisms such as β-catenin activation [46]. Further, increased MMP10 expression observed in androgen-independent cell lines as compared to androgen-dependent cell lines may be attributed to differential miR-410-mediated PRKD1 regulation in these cell lines. It has been shown that MMP10 expression is controlled by PRKD1 expression [47]. Increased MMP10 expression observed in androgen-independent cell lines upon miR-410 expression was correlated with increased in vitro invasiveness of these PCa cell lines. These data suggest that miR-410 likely impacts MMP10-mediated invasive behavior of PCa cell lines via its direct effects on PRKD1 expression.

EMT is a complex process influenced by multiple signaling pathways. Androgen-dependent and independent PCa cell lines might have distinct signaling pathway activities. We propose that miR-410 directly targets SNAIL in androgen-dependent cell lines that leads to EMT inhibition. However, in androgen-independent cell lines, miR-410 may be titrated away from its potential binding sites on SNAIL 3′ UTR likely via an alternate mechanism such as lncRNA (long non-coding RNA) binding. In androgen-independent cell lines, miR-410 was shown to regulate PRKD1 that leads to EMT induction. PRKD1 is known to cross-talk with AR signaling and interplay with β-catenin, leading to EMT induction [48,49]. These findings highlight the complexity and interplay of signaling pathways underlying prostate cancer EMT. It has been reported that in multiple myeloma cells, miR-410 is negatively modulated by lncRNA OIP5-AS1 that prevents its repression of its target Kruppel-Like factor 10 (KLF10) [50]. In agreement with our findings on the role of miR-410 in prostate cancer EMT, other miRNAs in this cluster including miR-154* and miR-379 were shown to facilitate tumor growth, EMT and prostate cancer bone metastasis [51]. PI3K-Akt signaling was found to be upregulated by miR-410 expression. These signaling pathways have been reported to be activated as cells undergo neuroendocrine differentiation upon AR targeted therapies [52]. In agreement, our data suggests that miR-410 upregulates synaptophysin and other neuronal markers. Future mechanistic studies will define the target genes directly regulated by miR-410 that promote therapy induced NED in PCa cells. GSEA analyses showed a potential regulatory role of miR-410 in controlling the expression of genes associated with the development and function of immune cells, particularly those involved in generating diverse immune receptors. Overall, our findings underscore the complexity of miRNA-mediated gene regulation and the dynamic nature of cellular processes in cancer. Further research focused on delineating miR-410 regulated signaling pathways in advanced PCa and its potential interplay with immune-related pathways is warranted.

In conclusion, our study defines a complex, context-dependent role of miR-410 in prostate cancer. Our data support the role of miR-410 in regulating the crucial process of EMT, thereby impacting PCa progression and metastasis. Our findings have important translational implications. In view of its critical roles in PCa, PRKD1 inhibition is exploited as a therapy for advanced prostate cancer [44]. In view of our data suggesting the potential inhibition of PRKD1 by miR-410, we suggest that miR-410 can be potentially exploited for advanced PCa therapy. Since miRNAs regulate pleiotropic targets, miRNA-based therapy is advantageous as compared to single agents.

## 5. Conclusions

Our research indicates that miR-410 serves as a tumor suppressor during the early phases of the disease by inhibiting epithelial-to-mesenchymal transition (EMT) through direct repression of the EMT-inducing transcription factor, SNAIL. However, in more advanced stages and higher grades of the disease, miR-410 adopts an oncogenic role via inhibition of PRKD1, possibly contributing to neuroendocrine differentiation. These discoveries hold significant significance in enhancing our comprehension of the molecular mechanisms underlying the progression of prostate cancer, potentially leading to translational applications.

## Figures and Tables

**Figure 1 cancers-16-00048-f001:**
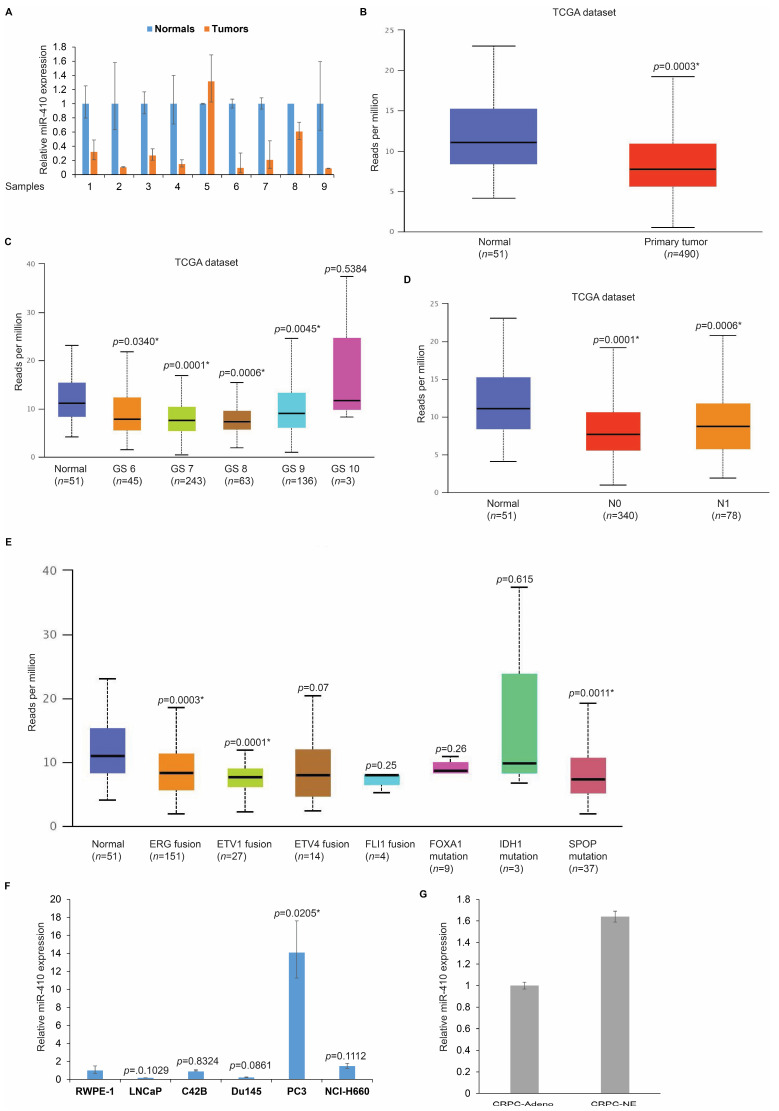
Analyses of miR-410 expression in PCa clinical samples and cell lines. (**A**) miR-410 expression analyses in matched tumor/normal PCa tissues (*n* = 9) as assessed by real-time PCR. RNU48 was used as an endogenous control. Error bars represent SEM. (**B**) miR-410 expression in TCGA dataset of prostate adenocarcinomas. Expression in normal tissues (*n* = 51) and primary prostate adenocarcinomas (*n* = 490) are represented. (**C**) miR-410 expression analyses in primary PCa stratified by Gleason score. (**D**) miR-410 expression in normal tissues vs. prostate adenocarcinomas with N0 or N1 tumor nodal status. (**E**) miR-410 expression in normal prostate tissues and various subtypes of prostate adenocarcinomas. (**F**) miR-410 expression analyses in normal immortalized prostate epithelial cell line RWPE-1 and indicated PCa cell lines. RNU48 was used as an endogenous control. (**G**) miR-410 expression in microdissected CRPC-Adenocarcinomas and CRPPC-NE clinical samples as assessed by real-time PCR analyses. RNU48 was used as an endogenous control. * denotes *p* < 0.05.

**Figure 2 cancers-16-00048-f002:**
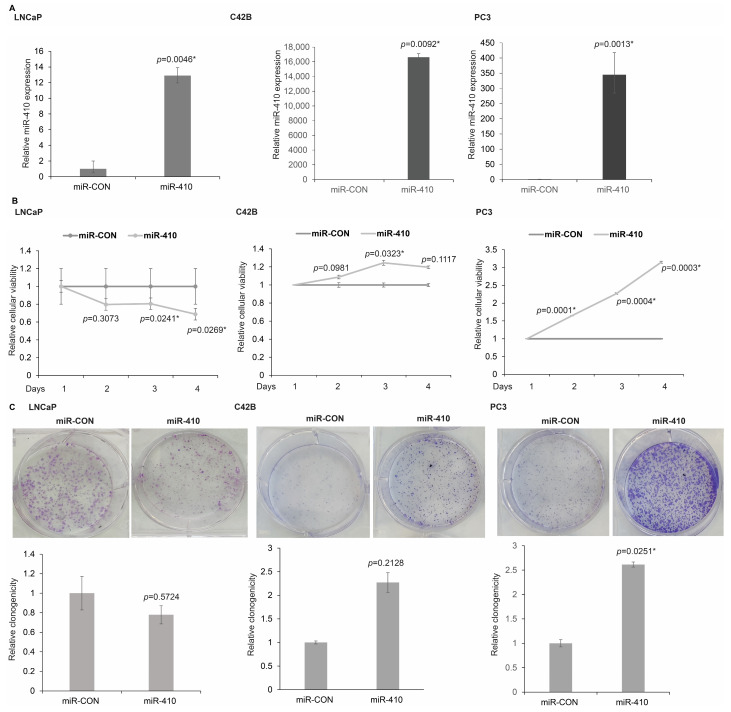
miR-410 regulates proliferation of PCa cells in a context-dependent manner. (**A**) Real-time PCR-based assessment of miR-410 expression in LNCaP, C42B and PC3 cell lines relative to their respective controls. (**B**) Relative cellular viabilities in LNCaP, C42B and PC3 cells lines transfected with miR-CON/miR-410 as assessed by MTS cellular viability assays. (**C**) Clonogenicity assays in miR-CON/miR-410 transfected LNCaP, C42B and PC3 cell lines. * denotes *p* < 0.05.

**Figure 3 cancers-16-00048-f003:**
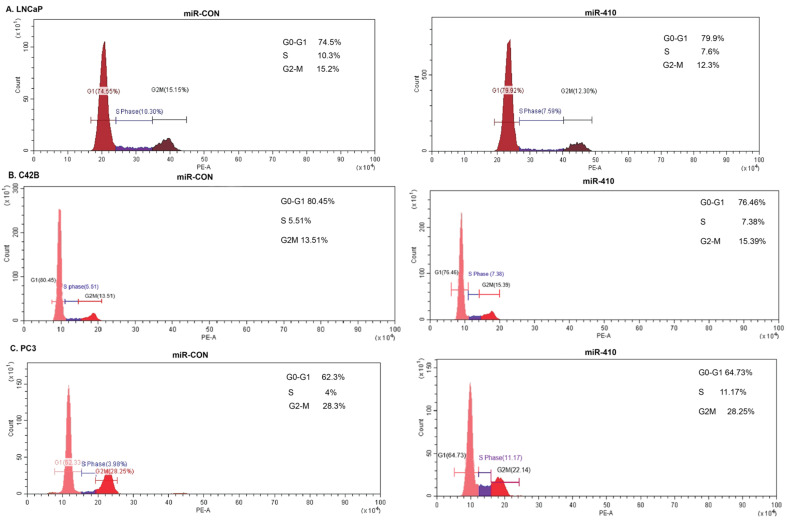
miR-410 regulates cell cycle progression of PCa cells in a context dependent manner. FACS analyses of propidium iodide stained (**A**) LNCaP, (**B**) C42B and (**C**) PC3 cell lines transfected with miR-CON/miR-410. Cell cycle analyses were performed after 72 h of transfection.

**Figure 4 cancers-16-00048-f004:**
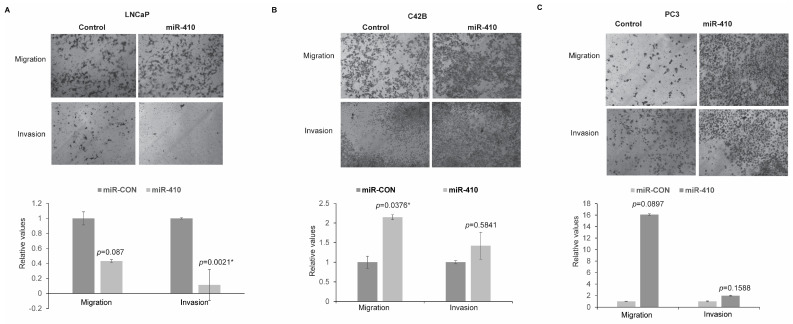
miR-410 overexpression influences migratory and invasive properties of prostate cancer cell lines. Transwell in vitro migration and invasion assay in control/miR-410 transfected (**A**) LNCaP, (**B**) C42B and (**C**) PC3 cells. Bar graphs below represent relative migratory/invasive abilities of transfected cells. Representative pictures of migrated/invaded cells are shown above. Pictures were taken on Keyence microscope (Itasca, IL, USA) at 10× magnification. * denotes *p* < 0.05.

**Figure 5 cancers-16-00048-f005:**
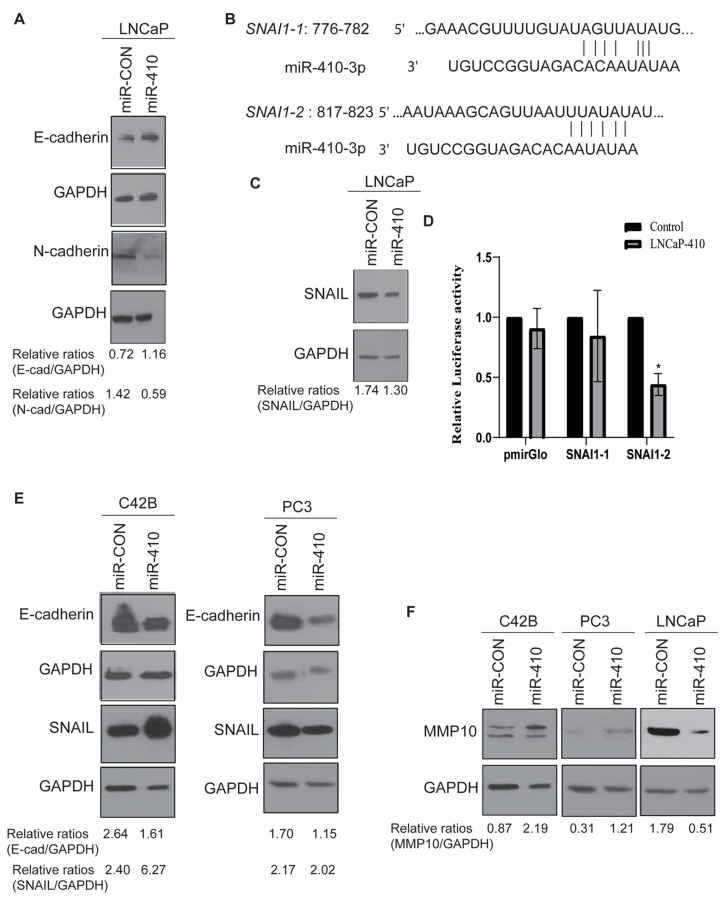
miR-410 regulates epithelial-to-mesenchymal transition in prostate cancer. (**A**) Western blot analyses of indicated proteins in miR-CON/miR-410 transfected LNCaP cells. GAPDH was used as a loading control. (**B**) Schematic representation of 3′ UTR of SNAI1 showing the complementarity of two putative binding sites, SNAI1-1 (Site 1) and SNAI1-2 (Site 2), with miR-410 seed sequence. (**C**) Western blot analyses of SNAIL expression in miR-CON/miR-410 transfected LNCaP cells. GAPDH was used as a loading control. (**D**) Luciferase reporter assay in LNCaP cells transfected with miR-CON/miR-410 and co-transfected with control pmiR-Glo or SNAI1-1 or SNAI1-2 3′ UTR constructs. Firefly and Renilla luciferase activities were measured, and the relative luciferase activities were plotted. (**E**) Western blot analyses of indicated proteins in miR-CON/miR-410 transfected PC3 and C42B cells. GAPDH was used as a loading control. (**F**) Western blot analyses of MMP10 in C42B, PC3 or LNCaP cells. GAPDH was used as a loading control. All Western blot images were analyzed by Image J Version 1.54d and relative band intensities were calculated and are shown below the blots. * denotes *p* < 0.05.

**Figure 6 cancers-16-00048-f006:**
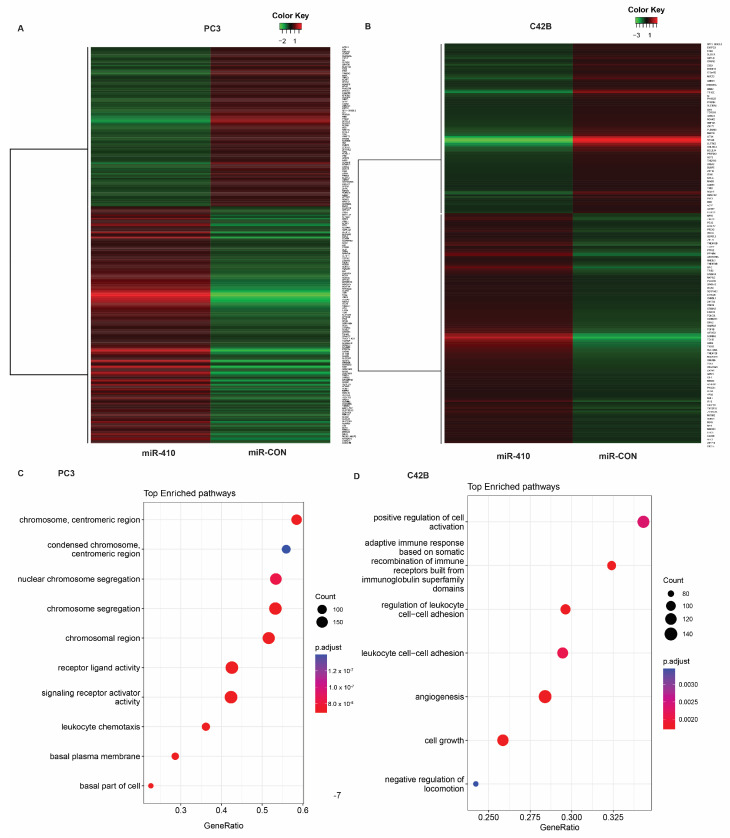
Microarray analyses of miR-410 target genes reveal its important role in critical cellular processes. (**A**) Heat map showing the dysregulated genes in control vs. miR-410 transfected PC3 cells; (**B**) heat map showing the dysregulated genes in control vs. miR-410 transfected C42B cells; (**C**) GSEA analyses showing top-enriched pathways in miR-410 transfected as compared to miR-CON transfected PC3 cells; (**D**) GSEA analyses of miR-410 transfected as compared to miR-CON transfected C42B cells.

**Figure 7 cancers-16-00048-f007:**
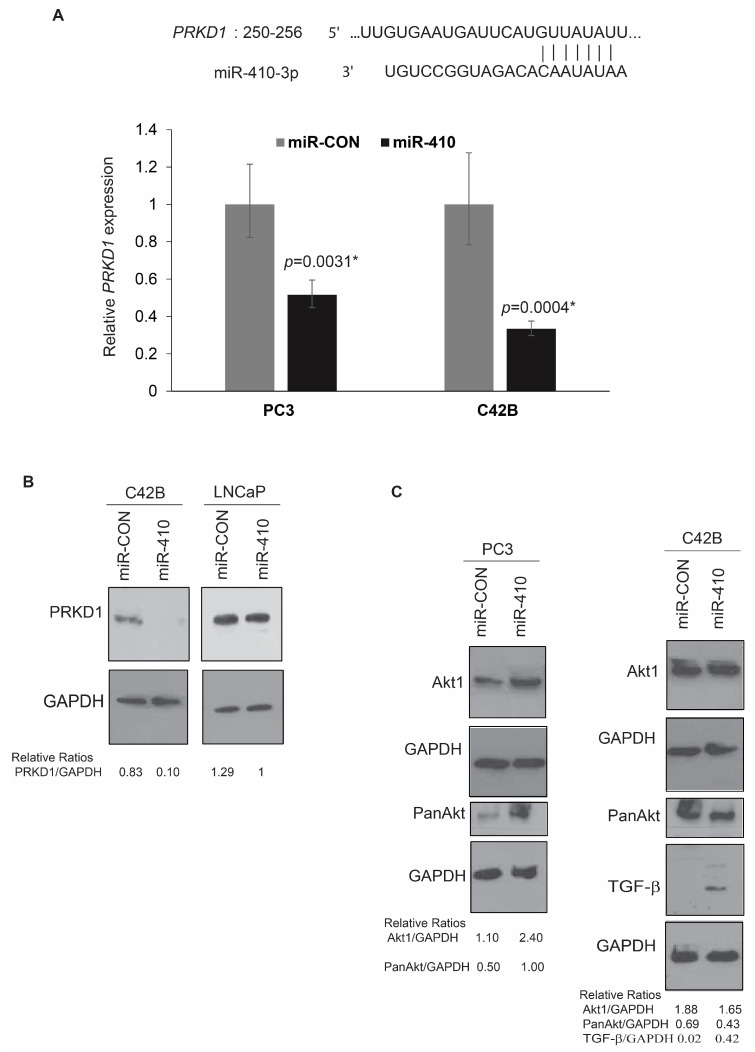
Protein Kinase D1 is a potential miR-410 target in prostate cancer. (**A**) **Upper** panel: Schematic representation showing potential miR-410 binding site in *PRKD1* 3’ UTR. **Lower** panel: Real time PCR analyses of relative PRKD1 expression in control vs miR-410 transfected PC3 and C42B cells. GAPDH was used as an endogenous control. (**B**) Western blot analyses of PRKD1 expression in indicated cell lines. GAPDH was used as a loading control. (**C**) Western blot analyses of indicated proteins in PC3 and C42B cells. GAPDH was used as a loading control. Band intensities were quantified by Image J and relative ratios are represented. * denotes *p* < 0.05.

**Figure 8 cancers-16-00048-f008:**
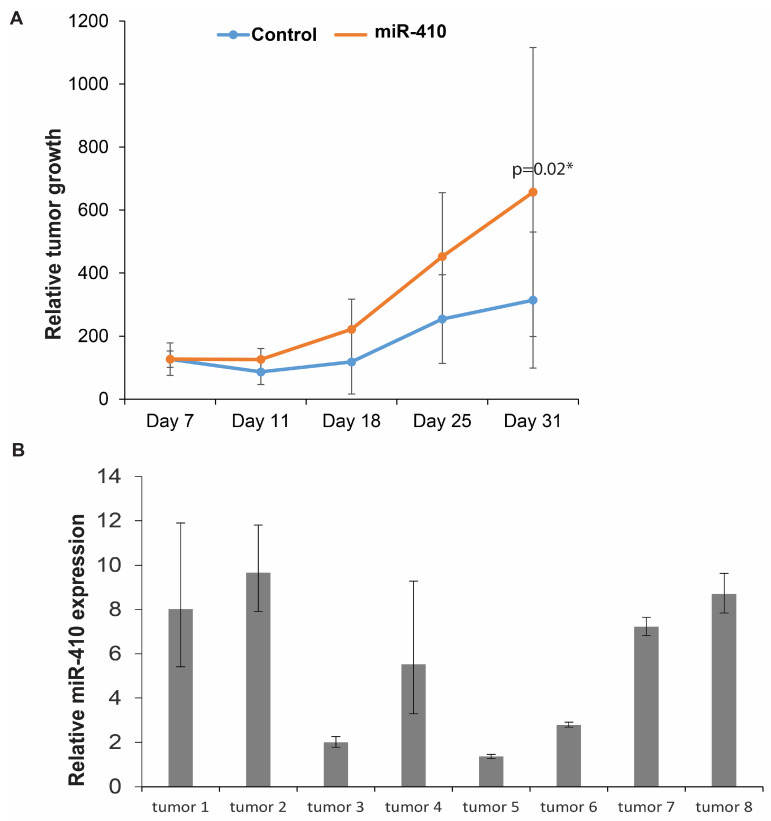
Effects of miR-410 overexpression in vivo. (**A**) Control miR or miR-410 expressing PC3 cells were subcutaneously injected into two groups of nude mice (*n* = 6 for control group, *n* = 8 for miR-410) followed by periodic monitoring of tumor growth. Relative tumor growths at various time points are shown. (**B**) Real-time PCR-based assessment of miR-410 expression in control vs. miR-410 expressing PC3 xenograft tumors. * denotes *p* < 0.05.

**Figure 9 cancers-16-00048-f009:**
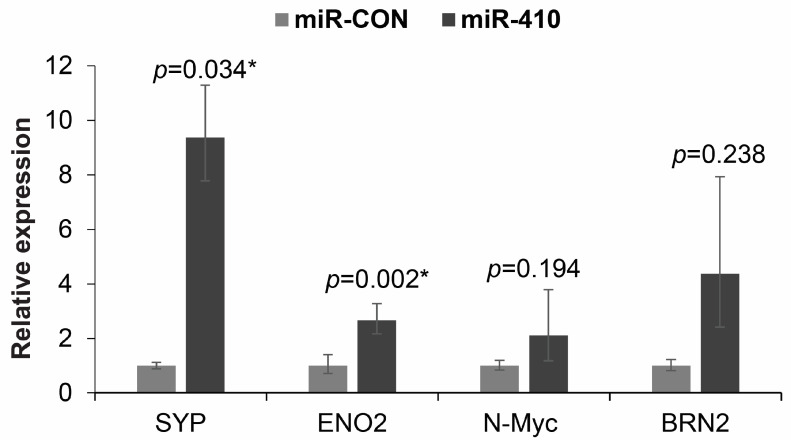
miR-410 regulates the expression of neuronal markers in prostate cancer. * denotes *p* < 0.05.

**Table 1 cancers-16-00048-t001:** Dysregulated genes identified by microarray analyses of miR-410 target genes in PC3 and C42B cell lines.

Cell Line	Gene	Fold Change
PC3	PRKD1	 −7.69474
	DPYSL3	 −7.63739
	C10orf107	 −6.0686
	FAM20C	 −6.02488
	ZNF385B	 −5.69932
	PDPN	 35.5498
	TC2N	 30.7099
	SLC6A14	 23.1731
	ESRP1	 22.6367
	CEACAM5	 20.9757
C42B	SLITRK3	 −88.5435
	CBLN2	 −51.0595
	BCHE	 −34.1878
	SPG20	 −16.0452
	COL12A1	 −11.7891
	IGFBP3	 24.7997
	TEX15	 12.3031
	SYT4	 11.8595
	CDH6	 11.082
	WLS	 7.63086


 Up-regulation; 

 Down-regulation.

## Data Availability

All reported data are included in the article. Microarray data generated in the study are deposited in Gene Expression Omnibus under Accession no. GSE245308.

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
