# Peer review of "miR-410 Is a Key Regulator of Epithelial-to-Mesenchymal Transition with Biphasic Role in Prostate Cancer"

_cancers, 2023, doi:10.3390/cancers16010048_

Round 1

Reviewer 1 Report

Comments and Suggestions for Authors

In this manuscript, the authors investigate the involvement of miR-410 in the progression of prostate cancer. The expression of miR-410 was analyzed in clinical samples and cell lines of prostate cancer (PCa). Functional studies of miR-410 were conducted in miRNA-control/miR-410 transfected LNCaP cells (early-stage epithelial cells), C42B, and PC3 cells (advanced-stage mesenchymal cells). Additionally, the authors explored the functional impact of miR-410 expression in vivo using a PCa xenograft mouse model.

The results indicate that miR-410 acts as a tumor suppressor in the initial stages of PCa (LNCaP), inhibiting epithelial-to-mesenchymal transition (EMT) through the direct repression of the EMT-inducing transcription factor, SNAIL. Conversely, miR-410 plays an oncogenic role in advanced PCa by promoting EMT and upregulating PI3K/Akt signaling in PC3 and C42B cells. In vivo studies with PC3 xenografts further support the oncogenic role of miR-410. This study unveils, for the first time, the biphasic role of miR-410 in PCa, demonstrating that its regulation of target genes is context-dependent.

To enhance the manuscript, addressing the following questions could provide further clarity:

1.      In result 3.3, it is demonstrated that miR-410 regulates cell cycle progression of prostate cancer cells in a context-dependent manner. However, there are issues with the clarity and overlap of labels in Figure 3, making it challenging to interpret the results effectively.

2.      In result 3.5, the study reveals that miR-410 plays a regulatory role in epithelial-to-mesenchymal transition (EMT) in prostate cancer. The authors conducted Western blot analysis on LNCaP cells, revealing an upregulation of the epithelial marker E-cadherin and a downregulation of the mesenchymal marker N-cadherin upon miR-410 expression compared to the corresponding control. However, in Figure 5A, there is no noticeable increase in E-cadherin expression observed in miR-410 cells compared to miR-CON. Similarly, in Figure 5E, there is no apparent difference in Akt1 expression in miR-CON/miR-410 transfected C42B cells. This suggests the need for quantification of the target bands in the Western blot to strengthen the conclusion. Moreover, it is notable that the presence of two target bands for MMP10 in C42B, in contrast to a single target band in PC3 and LNCap, raises intriguing observations. Similarly, in Figure 5E, the detection of two target bands for E-Cadherin in C42B, as opposed to the single target band observed in other cell lines, prompts the need for clarification. Providing an explanation for these results would enhance the overall clarity of the findings

3.      In result 3.6, microarray analysis of miRNA-410 target genes reveals its significant role in PI3-Akt signaling and EMT. To enhance clarity, the addition of a table displaying the top upregulated and downregulated genes is recommended. For Figure 6, it would be beneficial to group the upregulated and downregulated genes separately in the heatmap, providing a clearer visualization of the results. Additionally, including figures depicting the predominant dysregulation of the PI3K-Akt signaling pathway and other relevant pathways will further enhance the interpretation of the findings."

Author Response

We thank the reviewers for the constructive comments. We have addressed all the concerns raised by the reviewers as detailed point-by-point below

Comment 1: In result 3.3, it is demonstrated that miR-410 regulates cell cycle progression of prostate cancer cells in a context-dependent manner. However, there are issues with the clarity and overlap of labels in Figure 3, making it challenging to interpret the results effectively.

Response: As suggested, we have revised Figure 3 for clarity. Please refer to revised figure 3.

Comment 2:      In result 3.5, the study reveals that miR-410 plays a regulatory role in epithelial-to-mesenchymal transition (EMT) in prostate cancer. The authors conducted Western blot analysis on LNCaP cells, revealing an upregulation of the epithelial marker E-cadherin and a downregulation of the mesenchymal marker N-cadherin upon miR-410 expression compared to the corresponding control. However, in Figure 5A, there is no noticeable increase in E-cadherin expression observed in miR-410 cells compared to miR-CON. Similarly, in Figure 5E, there is no apparent difference in Akt1 expression in miR-CON/miR-410 transfected C42B cells. This suggests the need for quantification of the target bands in the Western blot to strengthen the conclusion. Moreover, it is notable that the presence of two target bands for MMP10 in C42B, in contrast to a single target band in PC3 and LNCaP, raises intriguing observations. Similarly, in Figure 5E, the detection of two target bands for E-Cadherin in C42B, as opposed to the single target band observed in other cell lines, prompts the need for clarification. Providing an explanation for these results would enhance the overall clarity of the findings.

Response: As suggested by the reviewer, we have addressed all these issues in revised Fig. 5. Please refer to results and discussion section of the revised manuscript (pages 12, 18-19). . We have quantified band intensities by Image J and calculated relative band intensities for each target analyzed (E-cadherin, SNAIL and/or N-Cadherin in Fig. 5; Akt now included in revised Fig. 7). The relative ratios are listed below the Western blots. Quantifications showed that E-cadherin expression is increased in LNCaP cells upon miR-410 expression. In contrast, its expression is decreased in PC3 and C42B. For Akt1, quantification showed an increase in PC3 cells and not C42B cells.

For C42B, we repeated Western blot analyses for E-cadherin expression and observed a single band. For MMP10, we observed two bands in C42B and PC3, likely corresponding to the precursor form and active form of MMP10. However, in LNCaP cells, a single band corresponding to the precursor form was observed. In addition, we provide new data to show that protein kinase D1 is a novel miR-410 target in PC3 and C42B cells (Fig.7). Increased MMP10 expression observed in androgen independent cell lines as compared to androgen dependent cell line may be attributed to differential miR-410 mediated PRKD1 regulation in these cell lines as it has been shown that MMP10 expression is controlled by PRKD1 expression.

Comment 3: In result 3.6, microarray analysis of miRNA-410 target genes reveals its significant role in PI3-Akt signaling and EMT. To enhance clarity, the addition of a table displaying the top upregulated and downregulated genes is recommended. For Figure 6, it would be beneficial to group the upregulated and downregulated genes separately in the heatmap, providing a clearer visualization of the results. Additionally, including figures depicting the predominant dysregulation of the PI3K-Akt signaling pathway and other relevant pathways will further enhance the interpretation of the findings."

Response: As suggested, we have now included a table (Table 1) showing the upregulated and downregulated genes upon miR-410 overexpression. We have revised Fig. 6 heatmaps as suggested. In addition, we provide GSEA analyses of the microarray data. Please refer to revised Fig. 6, Fig. S2-S3.

Reviewer 2 Report

Comments and Suggestions for Authors

The authors characterize the role for miR-410 in prostate cancer cell lines. To have this in context miR-410 is part of the largest microRNA family has been identified to be pro-tumorigenic in multiple cancer types regulating apoptosis, invasion, migration and angiogenesis. MiR-410 is also demonstrated to induce stemness. So, the findings shown here for prostate cancer are not surprising in that it promoted increased tumor growth and neuroendocrine differentiation by some cell lines. Of note, the authors performed microarray analysis of two cell lines (PC3 and C42B) over expressing miR-410. The authors state that omics analysis suggested the induction of EMT and PI3kinase signaling. Again, this is not surprising in light of the reports of GSK3ß activity with this microRNA. However, it is necessary to show the analysis performed and the ensuing results from this analysis. For example, GSEA plots, master regulator analysis, and indications of significance of the findings. The heatmap shown in Figure 6 is unlabeled. It would be important to validate some of the genes identified in the microarray in some fashion. At the least, the tumor tissue can be easily measured for the expression of the PI3K and EMT genes.

Other issues that require attention:

1)    The authors chose to use C42B and PC3 lines for many of the studies. when considering EMT and neuroendocrine features, the parental PC3 cells are already neuroendocrine and express many EMT genes (and even morphology). Thus, they saw no change in neuroendocrine genes in Figure 8B (although stated otherwise). So, it does not make sense to look at synaptophysin expression in the PC3 xenograft tumor tissues. The bar graph is not well described or labeled – suggest it be removed. Better served to measure the changes seen in the microarray.

2)    A better background on the known work with miR-410 should be provided so that the new findings here can be put into context. For example, the results in figure 5E, where AKT levels were found to be higher with miR-410 expression makes sense, as GSK3 was previously found to be elevated with this microRNA.

3)    What is the significance of the MMP10 regulatory function of miR-410? It seems to be differentially regulated by the 3 cell lines tested. This may be worth exploring further.

Author Response

Comment 1: The authors characterize the role for miR-410 in prostate cancer cell lines. To have this in context miR-410 is part of the largest microRNA family has been identified to be pro-tumorigenic in multiple cancer types regulating apoptosis, invasion, migration and angiogenesis. MiR-410 is also demonstrated to induce stemness. So, the findings shown here for prostate cancer are not surprising in that it promoted increased tumor growth and neuroendocrine differentiation by some cell lines. Of note, the authors performed microarray analysis of two cell lines (PC3 and C42B) over expressing miR-410. The authors state that omics analysis suggested the induction of EMT and PI3kinase signaling. Again, this is not surprising in light of the reports of GSK3ß activity with this microRNA. However, it is necessary to show the analysis performed and the ensuing results from this analysis. For example, GSEA plots, master regulator analysis, and indications of significance of the findings. The heatmap shown in Figure 6 is unlabeled. It would be important to validate some of the genes identified in the microarray in some fashion. At the least, the tumor tissue can be easily measured for the expression of the PI3K and EMT genes.

Response: As suggested, we have extensively revised the manuscript to include additional data. We now provide new data from omics analyses including GSEA analyses (Fig. 6 and Fig. S2-S3) and validation of microarray data (revised Fig. 7). We validate one of the target genes, Protein Kinase D1 (PRKD1) as a potential miR-410 target. We did not have tumor tissues from in vivo studies. Therefore, we validated the data in PC3 and C42B cell lines. We have revised the discussion to include significance of the findings. Please refer to pages 14-16, 19-20 of the revised manuscript. We have revised the heatmaps in Fig. 6 to include labels. 

Comment 2:    The authors chose to use C42B and PC3 lines for many of the studies. when considering EMT and neuroendocrine features, the parental PC3 cells are already neuroendocrine and express many EMT genes (and even morphology). Thus, they saw no change in neuroendocrine genes in Figure 8B (although stated otherwise). So, it does not make sense to look at synaptophysin expression in the PC3 xenograft tumor tissues. The bar graph is not well described or labeled – suggest it be removed. Better served to measure the changes seen in the microarray.

Response: As suggested, we have now removed the expression of NE genes in PC3 cell line and xenografts in Fig. 9 (earlier Fig. 8). We measured changes in the microarray and importantly, we identify PRKD1 as a novel miR-410 target gene in androgen independent cell lines (revised Fig. 7). PRKD1, belongs to the family of calcium calmodulin kinases, that has been reported to be reduced in metastatic prostate cancer as compared to localized disease. PRKD1 signaling plays an important role in the regulation of cellular proliferation, adhesion, EMT, invasion, migration, immune response and angiogenesis. Interestingly, it has been reported earlier that PRKD1 suppresses EMT through phosphorylation of SNAIL. PRKD1 phosphorylates SNAIL on Serine 11 that triggers nuclear export of SNAIL via binding to a 14-3-3s protein. Upon nuclear export, phosphorylated SNAIL cannot repress E-cadherin effectively, leading to E-cadherin induction. In light of these studies and our present results, we propose that the biphasic expression of miR-410 expression in PCa cell lines is inversely correlated to PRKD1 expression that determines PCa EMT outcomes. We propose that in advanced prostate cancer, miR-410 upregulation leads to PRKD1 inhibition that promotes EMT, likely via non-phosphorylation of SNAIL and other mechanisms such as b-catenin activation. We have included these new findings in the results and discussion section.

Comment 3: A better background on the known work with miR-410 should be provided so that the new findings here can be put into context. For example, the results in figure 5E, where AKT levels were found to be higher with miR-410 expression makes sense, as GSK3 was previously found to be elevated with this microRNA.

Response: We have now included a detailed background on miR-410 in revised introduction. Please refer to pages 1-2 (lines 96-99) of the revised manuscript.

Comment 4: What is the significance of the MMP10 regulatory function of miR-410? It seems to be differentially regulated by the 3 cell lines tested. This may be worth exploring further.

Response: We agree with the reviewer. We found that miR-410 regulates PRKD1. Increased MMP10 expression observed in androgen independent cell lines as compared to androgen dependent cell line may be attributed to differential miR-410 mediated PRKD1 regulation in these cell lines as it has been shown that MMP10 expression is controlled by PRKD1 expression. Increased MMP10 expression observed in androgen independent cell lines upon miR-410 expression was correlated to increase in in vitro invasiveness of these PCa cell lines. These data suggest that miR-410 likely impact MMP10 mediated invasive behavior of PCa cell lines via its direct effects on PRKD1 expression. These points are included in discussion section (page 20, lines 543-548).

Round 2

Reviewer 1 Report

Comments and Suggestions for Authors

The authors have addressed my inquiries. Kindly review the entire manuscript, correcting any typos, and adjusting the formats of the figure labels

Reviewer 2 Report

Comments and Suggestions for Authors

The authors have made extensive edits to the manuscript inclusive of the addition of new interesting data.